# How Different Snacks Produce a Distinct Effect in Salivary Protein Composition

**DOI:** 10.3390/molecules26092403

**Published:** 2021-04-21

**Authors:** Carla Simões, Inês Caeiro, Laura Carreira, Fernando Capela e Silva, Elsa Lamy

**Affiliations:** 1Mediterranean Institute for Agricultural, Environmental and Development, University of Évora, 7002-556 Évora, Portugal; carlasimoes3@hotmail.com (C.S.); ines_caeiro_@hotmail.com (I.C.); lrec@uevora.pt (L.C.); 2Department of Medical and Health Sciences, School of Health and Human Development and Mediterranean Institute for Agricultural, Environmental and Development, University of Évora, 7002-556 Évora, Portugal; fcs@uevora.pt

**Keywords:** alpha-amylase, cystatins, food intake, immunoglobulins, salivary proteins

## Abstract

Saliva secretion changes in response to different stimulation. Studies performed in animals and humans suggest that dietary constituents may influence saliva composition, although the dynamics of these changes, and how they are specific for each type of food, are little known. The objective of the present study was to access the short-term effects of different foods in salivation and salivary protein composition. Twelve participants were tested for four snacks (yoghurt, bread, apple and walnuts). Non-stimulated saliva was collected before and at 0′, 5′ and 30′ after each snack intake. Flow rate, total protein, alpha-amylase enzymatic activity and salivary protein profile were analyzed. Yoghurt and apple were the snacks resulting in higher salivary changes, with higher increases in flow rate and alpha-amylase activity immediately after intake. The expression levels of immunoglobulin chains decreased after the intake of all snacks, whereas cystatins and one pink band (proline-rich proteins—PRPs) increased only after yoghurt intake. Walnut’s snack was the one resulting in lower changes, probably due to lower amounts eaten. Even so, it resulted in the increase in one PRPs band. In conclusion, changes in saliva composition varies with foods, with variable changes in proteins related to oral food processing and perception.

## 1. Introduction

Saliva is a biological fluid, on which attention has been growing in the last years. One of the reasons is the potential of this fluid as a non-invasive source of biomarkers for several pathological and physiological states [1], since many of the molecules existent in blood are also present in saliva. Moreover, the role of this fluid in oral and systemic health is recognized [1] and the participation of saliva in oral food perception has also started gaining interest [2]. One of the characteristics of saliva is its plasticity, with a quick and strong response to different types of stimuli, changing in volume and composition. It is the fact of being mainly regulated by the autonomic nervous system that confers this characteristic to saliva [3], with variable changes in response to different external and internal factors, such as stress, time of the day or even dietary habits [4].

The possibility of saliva being influenced by dietary characteristics is recognized in studies made with animals that demonstrate differences in saliva composition among species that have different feeding habits [5], as well as differences among genes that codify for salivary proteins in species belonging to different feeding niches [6].

In humans, the first studies that related dietary constituents with saliva had astringency as the base. Astringency was the first oral food sensation for which the participation of salivary proteins was recognized (e.g., [7,8]), and only after the association between saliva composition and basic tastes sensitivity was considered [9,10,11,12,13]. In the particular case of astringency, some particular families of salivary proteins have been reported as participating in this sensation, due to their high affinity to complex polyphenols [14]. This is the case of the proline-rich proteins (PRPs) family, particularly the basic PRPs, for which the only recognized function was to bind these plant compounds. At the same time that salivary proteins were reported to influence astringency derived by polyphenols (tannins), it was observed that diets rich in these compounds induced the synthesis and production of PRPs and other salivary proteins in rodents, like mice or rats [15,16]. This reinforced the possibility of an effect of diet composition in saliva composition, but in humans, the number of studies assessing salivary response to dietary tannins/polyphenols is lower, although there are reports of short-term effects in the salivary protein secretion induced by basic solutions of tannic acid [8], cranberries polyphenols [17] and chocolate milk polyphenols [18].

Variations in salivary protein composition were also observed in sequence of long-term high-fat diet treatment [19], or even after bread chewing [20,21], reinforcing the hypothesis that saliva responds to ingestion and changes accordingly.

The short-term effect of different snacks in saliva pH was previously reported [22], from where it became evident that short-time responses to food intake are not the same for foods with different composition. However, a potential effect in other parameters, besides pH, was not assessed.

Considering the known role that salivary proteins have in food processing and in sensory perception (e.g., [12,13,23]), it is possible to think that the changes in salivary protein composition, induced by food intake, can have practical effects in food acceptance and preference and even in digestive/metabolic processes. In fact, it was observed, using animal models, that salivary protein profiles influenced food sensory acceptance [24] and, in humans, that salivary proteome is associated with sensory ratings [21].

Given the importance salivary changes can have for oral food processing, perception and acceptance, as well as for oral health or even to access specific biomarkers, more detailed knowledge about the way this fluid responds to different foods is of relevance. As such, the objective of the present work was to study the effect of different types of snacks in salivary protein composition. Our work hypothesis is that foods with different compositions and/or sensory characteristics result in specific short-term changes in salivary protein profile.

## 2. Results

### 2.1. Flow Rate, Total Protein Concentration and Alpha-Amylase Enzymatic Activity

The intake of the different snacks resulted in changes in saliva secretion and composition, which appeared to be snack specific. Yoghurt was the snack producing higher increase in flow rate, followed by apple (Figure 1A). Bread and walnuts did not induce a statistically significant increase in saliva flow rate (Figure 1A). Concomitant with flow rate increase, total protein concentration decreased immediately after yoghurt intake, continuing to decrease 5 min after intake, being totally recovered when saliva was collected 30 min after ingestion. On the contrary, and despite the effect of apple in increasing flow rate, this snack resulted in increases in total protein concentration, which returned to control values 5 min after ingestion. For bread and walnuts intake, no significant changes were observed at protein concentration level (Figure 1B).

The enzymatic activity of alpha-amylase also varied in different ways according to the type of snack. It increased significantly immediately after yoghurt and apple intake, returning to values like the ones before intake, at 5 min. Walnut’s ingestion did not change the activity levels of this enzyme (Figure 1C). Results from two-way ANOVA analysis are presented in Table 1.

### 2.2. Salivary Protein Profile

SDS PAGE protein separation allowed the observation of 11 bands, consistently present in the several profiles (Figure 2). The tryptic digestion, followed by mass spectrometry, resulted in the identification of 8 different proteins (Table 2).

The expression levels of some of these proteins were observed to change after snacks intake, with the type of snack influencing the changes (Table 1). The bands containing chains of immunoglobulins (band A and band G) decreased in the expression levels immediately after the ingestion of the snacks. Exceptions were the cases of walnuts, where the change in band A was not significant, and bread, where the change in band G was not significant (Figure 3).

The bands containing cystatins (bands J and K) increased immediately after intake of all snacks, except walnuts (Figure 4).

Band B, containing albumin, presented different variations according to the type of snack: for apple it increased immediately after intake, remaining at higher levels, at least, 5 min after, whereas after yoghurt intake, the expression levels of this band decreased, being already at control levels 5 min after intake.

The staining protocol used for the gels allowed the observation of bands with violet/pink coloration, which are expected to be proline-rich proteins (PRPs) [25], in the regions of 75–100, 35 and 23 kDa (Figure 5). These last two bands correspond to bands F and H, which were not identified by mass spectrometry. Since trypsin fails in cleave the polypeptide chain when lysine or arginine are close to a proline, PRPs usually fail MS identification, reinforcing the hypothesis of these two bands corresponding to PRPs. In the case of band F, it increased after yoghurt intake, being at control levels 5 min after, whereas band H increased only after walnuts ingestion, remaining higher for at least 5 min. The evolution of these two bands, through time, can be seen in Figure 3.

## 3. Discussion

Since Pavlov’s studies [26], it is accepted that salivary secretion changes in response to food stimulation. What is less well known is how these changes process (only increased volume or also changed composition) and if the type of response is equal, independently of the type of food consumed, or rather, specific to food type. In the present study, four different types of foods, usually eaten as snacks, were studied for their short-term effect in salivary protein composition. To control for potential effects of energy level, the quantity of each snack consumed was adjusted to get equal energy content. Moreover, since different factors can affect saliva composition, such as sex and BMI [13], as well as smoking and alcohol habits [27], only normal-weight non-smoker and non-alcohol consumers were included.

As hypothesized, ingesting foods with different composition and sensory characteristics resulted in different effects in salivation. A statistically significant increase in flow rate was only observed for yoghurt and apple snacks. These two snacks have in common the fact of being sweet-sour [28] and it is known that sourness stimulates saliva secretion [29]. But if, on one hand, both induced significant increases in the flow rate, only apple induced a simultaneous increase in total protein concentration. Since it is not possible to know if this results from other sensory stimulation and/or difference in compounds that are specific of apple or yoghurt, the hypothesis is that a higher masticatory requirement for apple can contribute to this increased salivary protein concentration. No consensus exists about the effect of mastication (force and/or duration) on saliva total protein secretion [30]. Chewing non-taste materials, like parafilm, resulted in decreases in total protein concentration [31], whereas a study performed with food chewing (bread) observed increased total protein concentration [32]. Interestingly, this last study evaluated bread and showed different effects in total protein concentration induced by different types of breads, with industrial bread having a minor effect, comparative to whole meal or artisan bread. This may explain our results, where a wheat bread, probably closer to industrial bread, was tested. Moreover, previous studies from our team, using a similar type of bread, also failed to find increases in total protein concentration after bread chewing [20,21]. This highlights the different effects, in total protein concentration, of foods with different composition. It is important to highlight the high variability in salivary total protein concentration values, which explains why the apparent variation through time, observed in Figure 1B, resulted in no statistically significant differences.

Alpha amylase is one of the salivary proteins present in higher levels in saliva, being mainly secreted by parotid glands. Whereas bread did not result in significant changes in the levels of this protein, neither in terms of enzymatic activity nor expression levels, yoghurt and apple intake significantly increased the amylolytic activity of saliva. The lack of significant effect of bread chewing/intake in salivary amylase activity is in line with previous studies [20,21,32]. However, the effect of yoghurt and apple, in salivary amylase activity, is, to our knowledge, first reported in the present study. The increase due to yoghurt intake can be not a real increase in the secretion of the salivary protein, as the expression levels of the bands containing this protein did not present changes, but rather a collateral effect due to the high content of calcium in yoghurt. In fact, an enhancement of salivary amylase activity by the addition of calcium, at 37 °C, was reported [33]. Since saliva collected immediately after yoghurt intake can contain residuals of yoghurt (although mouth was cleaned with water), and consequently of calcium, this hypothesis needs to be considered. Taking into consideration the role of amylase activity in in-mouth sensory perception, this effect can have consequence in the oral perception of foods consumed together or right after yoghurt intake.

With the exception of walnuts, all the other types of snacks resulted in decreases in the levels of immunoglobulin chains. Similarly, in other studies, a decrease in the expression levels of these proteins were also observed after the mastication of bread or rice [20,21]. As well, other authors observed decreased salivary immunoglobulin levels induced by the mechanical plus gustatory stimulation, characteristic of chewing [34,35]. Altogether, this suggests that mastication of different types of foods can induce decreases in salivary immunoglobulin levels.

Yoghurt was the snack producing more changes in salivary protein profile. Besides the already reported ones, the expression levels of the bands containing S-type cystatins increased, as well as band F, which, although being not identified by mass spectrometry, stained pink after Beeley’s coloration for PRPs [25]. According to this coloration and the molecular mass, it is expected band F contain basic PRP1 (UNIPROT P04280). Both cystatins and basic PRPs have been linked to astringency. Salivary PRPs are considered the leading family of salivary proteins associated with astringency [36], with cystatins being also a class of salivary proteins associated with the perception of this oral tactile sensation [8,37]. Whereas the majority of studies on astringency have been done in polyphenol (tannin) rich products, this oral tactile sensation has been also reported in yoghurts [38]. Although the mechanism through which dairy proteins (and particularly whey proteins) result in astringency is not fully elucidated, interactions between whey proteins and salivary PRPs and mucins have been proposed (reviewed in [39]). The role of cystatins in these processes is not clear, but taking into account some common aspects between astringency induced by polyphenols and whey proteins, it is possible to hypothesize that the increased levels of salivary cystatins and PRPs resultant from yoghurt can be related with this sensory aspect of this snack.

Band H, another pink protein band, increased only after walnuts intake. Walnuts contain polyphenols, with variable amounts of tannins, which are positively related with astringency [40]. In line with the thought of astringency effect in the secretion of PRPs, it is possible that this may be the cause of this increase. However, further studies need to be done to test this hypothesis. Moreover, this was the only protein band significantly changed after walnuts intake. The lower amount of this snack, comparative to the other snacks tested, in order to have iso-energetic snacks, resulted in lower chewing and lower contact of walnuts with oral tissues and saliva, which can explain the lower effects in salivation.

## 4. Materials and Methods

### 4.1. Participants

A total of twelve healthy women aged between 18–30 years took part in this study. The choice of participants from only one sex in this study was due to minimizing differences in saliva composition regarding potential sex influence in this fluid composition. The exclusion criteria were the presence of oral or systemic diseases, the use of medication and smoke or alcohol consumption habits. Moreover, reported Body Mass Index (BMI) was considered, with only normal-weight women (18 < BMI < 25 kgm^−2^) participating.

The study was performed in accordance with the ethical guidelines for scientific research and approved by the Ethical Committee of the University of Evora (GD/2746/2021).

### 4.2. Experimental Design and Food Stimulus

This study was conducted at the laboratory settings of the University of Evora, following a design where all the participants were exposed to the four different snacks in study. To avoid a potential cumulative effect of previous snack in the subsequent one, the experiment occurred in 4 sessions, with 7 days interval between each of them, between February and March 2019. In the first session, subjects were randomly divided in 4 groups (3 subjects per group) and each group were randomly assigned for one of the snacks. The participants tested snacks A, B, C and D in the first section, tested snacks B, C, D and A, respectively, in the second session, snacks C, D, A and B, respectively, in the third session and snacks D, A, B and C, respectively, in the fourth session. This allowed that all the individuals went through all the stimuli in study and all snacks were tested in each of the session’s days. In order to have all participants in the same condition at the moment of the tests, in each session, participants were asked to come to the laboratory in fasting condition, between 09:00 and 09:30 am, the breakfast being served at 9.30 am for all participants. The breakfast was equal for all participants, in all session days. It consisted of a butter sandwich (40 g wheat bread), 200 mL of chocolate soy drink (Alpro Soya, Ghent, Belgium) and an express coffee. Coffee was considered because all participants referred to consumed it daily, after breakfast. Participants were instructed to consume the totality of the food provided. Time of ending was registered, and participants were instructed to be in a calm place, performing activities like reading or computer working, for 90 min. After that time, the snacks experiment began. Saliva samples were collected in four different occasions: 90 min after the breakfast and immediately before snack consumption (Before); immediately after snack ingestion (0′); 5 min after snack ingestion (5′); 30 min after snack ingestion (30′).

The snacks used in the present study were apple (Royal Gala apple, 180 g); bread (wheat white bread, 48.4 g); tutti-fruit aroma yoghurt (125 g); walnuts (15.7 g), in order to be sensorially different, representative of the usual snack choices and providing the same amount of energy. The aroma yoghurts used in the different sessions were from the same commercial brand, to avoid differences due to different fabricants.

### 4.3. Saliva Collection and Laboratory Analysis

Saliva was collected in the absence of stimulation. Individuals were instructed to drink water to eliminate any residual saliva and to wait 30 s, after which they did not swallow, for 4 min, spiting all the saliva produced in the mouth into a clean polyethylene tube maintained in ice. Immediately after collection, tubes were weighed and stored at –28 °C, until laboratory analysis. In the days after collection, saliva samples were thawed on ice and centrifuged at 13,000× *g* 4 °C for 20 min, to precipitate insoluble material and recover homogeneous liquid samples.

#### 4.3.1. Salivary Flow Rate, Total Protein Concentration and Alpha Amylase Activity

Saliva flow rate was assessed by assuming that saliva density is 1.0. The weight of empty tubes was subtracted from the weight of the tubes containing saliva and the final value was divided by 4 (minutes of collection) to obtain the secretion rate (mL/min). Total protein concentration was determined by the Bradford method, using bovine serum albumin (BSA) as standard, and plates were read at 600 nm in a microplate reader (Glomax, Promega, Madison, WI, USA).

For salivary amylase enzymatic activity quantification, a Salimetrics^®^ kit was used according to the manufacturer’s recommendations, as previously described and using saliva samples diluted 200X [20]. Absorbance values were read at 405 nm in a plate reader spectrophotometer (BioRad, Hercules, CA, USA), at two time points and the enzymatic activity of amylase (U/mL) was calculated.

#### 4.3.2. SDS-PAGE Salivary Protein Separation

Each saliva sample was run in duplicate. For each sample, a volume corresponding to 6.5 µg total protein was mixed with sample buffer and run on each lane of a 14% polyacrylamide mini-gel (Protean xi, Bio-Rad, CA, USA) using a Laemmli buffer system, as described elsewhere [19]. An electrophoretic run was performed at a constant voltage of 140 V until front dye reached the end of the gel. Gels were fixed for 1 h in 40% methanol/10% acetic acid, followed by staining for 2 h with Coomassie Brilliant Blue (CBB) G-250. Gel images were acquired using a scanning Molecular Dynamics densitometer with internal calibration and LabScan software (GE Healthcare, Chicago, IL, USA), and images were analyzed using GelAnalyzer software (GelAnalyzer 2010a by Istvan Lazar, www.gelanalyzer.com, assessed on February 2020) for the normalized volume (volume percentage) of each protein band. Molecular masses were determined in accordance with molecular mass standards (Bio-Rad Precision Plus Protein Dual Colour 161–0394) run with protein samples.

#### 4.3.3. Protein Identification by Mass Spectrometry

Bands of interest were manually excised from gels and proteins were in-gel digested following a protocol previously described [13].

To identify target proteins, peptide mixtures were analyzed by MALDI- FTICR-MS in a Bruker Apex Ultra, Apollo II combi-source (Bruker Daltonics, Bremen, Germany), with a 7 Tesla magnet (Magnex corpora- tion, Oxford, UK). After samples were desalted and concentrated, using reverse phase Poros R2 (Applied Biosystems, Foster City, CA, USA), they were eluted directly to the MALDI target AnchorChip (BrukerDaltonics, Bremen, Germany) with a-cyano-4-hydroxycinnamic acid (CHCA; Fluka, Buchs, Switzerland) matrix, prepared at a concentration of 10 lg/ll in 50% ACN with 0.1% TFA. Monoisotopic peptide masses were determined using the SNAP 2 algorithm in Data Analysis software version 3.4 (BrukerDalton- ics, Bremen, Germany). External calibration was performed using the BSA tryptic digest spectrum, processed and analyzed with Biotools 3.1 (BrukerDaltonics, Bremen, Germany).

Monoisotopic peptide masses were used to search for protein identification with Mascot software (Matrix Science, London, UK), in the Swiss-Prot non-redundant protein sequence database, restricted to Homo Sapiens. A minimum mass accuracy of 10 ppm, one missed cleavage in peptide masses, carbamido-methylation of Cys and oxidation of Met, as fixed and variable amino acid modifications, respectively, were considered. Criteria used to accept the identification were homology scores higher than 56 achieved in Mascot.

### 4.4. Statistical Analysis

Descriptive statistics was performed, and data normal distribution and homoscedasticity were tested through Shapiro–Wilk and Levene tests, respectively. To evaluate the differences between snacks, in terms of the changes in salivary parameters produced through time, a within subjects’ two-way ANOVA repeated measures analysis. The 4 snacks and the 4 collection periods (R1, R2, R3 and R4) were considered as the 2 factors. A Bonferroni-type adjustment was made to prevent alpha inflation.

All these statistical procedures were performed for saliva flow rate, total protein concentration, salivary amylase enzymatic activity and normalized spot volume. Statistical analysis was performed using SPSS v. 24, with significance level set at 5%.

## 5. Conclusions

Here we present preliminary results showing that the type of food eaten is not indifferent in the type of salivary changes produced. Eating different iso-energetic snacks results in changes in the relative amounts of different types of salivary proteins. Yoghurt and apple were the snacks inducing higher changes in salivary response what can be due to the sensory physical or chemical properties of these foods. Walnuts, on the other hand, resulted in small changes, probably due to the low amount eaten to reach the same energy content. It is important to highlight that most of the immediate changes are no longer observed after 5 min post-intake, although some salivary parameters’ changes can remain after that time. The effect of these changes in oral food perception of subsequent eaten foods was beyond the scope of this study and was not accessed, but among the proteins observed to be changed are salivary proteins previously observed to affect oral food perception, with particular relevance for proteins involved in astringency. As such, further studies, assessing the changes in sensory perception, can be of interest in the sensory area, to understand and develop well accepted food combinations, or even in health, since sensory perception can affect food intake, affecting health outcomes, such as obesity development.

This study has some limitations. One of them is the limited number of participants. The methodological approach, namely, the proteomics gel-based approach, limits the number of samples, even more since each sample was run in duplicate to minimize error due to gel-to-gel variations. Nevertheless, the experimental design was conceived to allow to have each individual as its own control, which is an advantage taking into account the inter-individual variation commonly found in saliva proteome. Moreover, only women were tested, to minimize variability due to potential sex differences, and all individuals had the same meal before the experiment, to minimize variations due to an effect of the type of food previously ingested. In the present study, only one-dimensional electrophoresis was used, limiting the number of changes able to be observed at salivary protein level. Mucins, which are particularly relevant in salivary lubrification, for example, could not be studied using this approach.

## Figures and Tables

**Figure 1 molecules-26-02403-f001:**
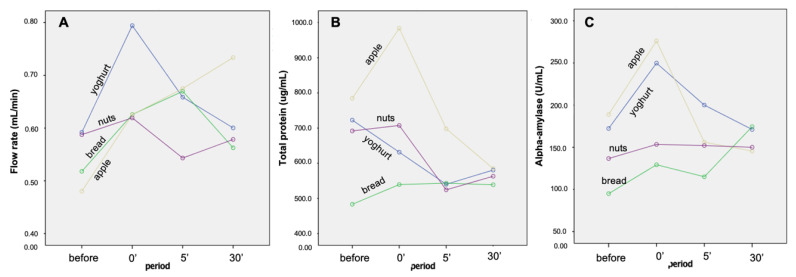
Variations of flow rate (**A**), total protein concentration (**B**) and salivary alpha-amylase enzymatic activity (**C**) induced by intake of different snacks.

**Figure 2 molecules-26-02403-f002:**
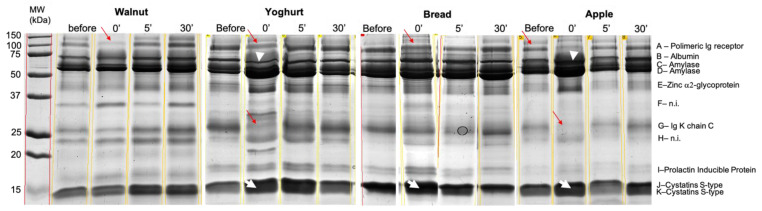
SDS-PAGE profiles representative of saliva collected before and after the ingestion of the four snacks studied. Bands A and G decreased immediately after ingestion (0’) of snacks (red arrow); Band B decreased immediately after yoghurt and increased immediately after apple intake (white arrowhead); Bands J + K increased immediately after the intake of all foods except walnuts (white arrows).

**Figure 3 molecules-26-02403-f003:**
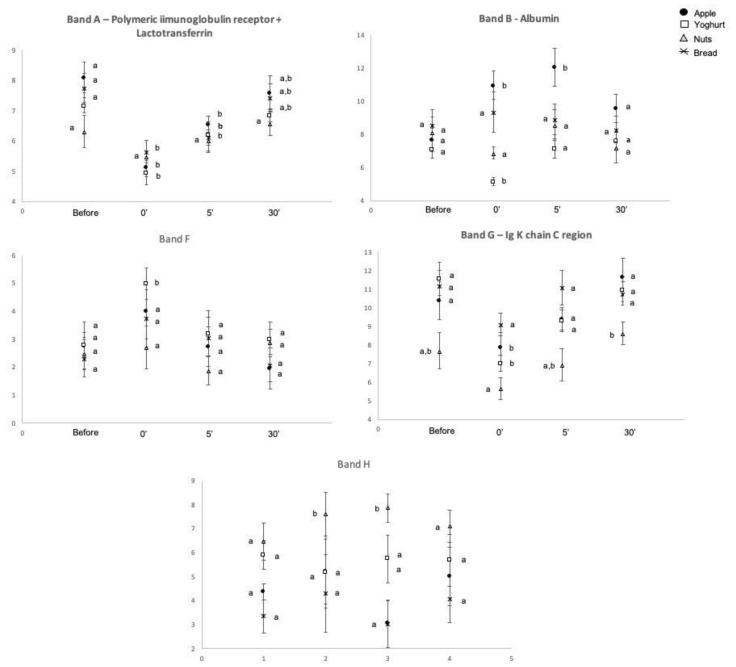
Variations in salivary proteins induced by the intake of different snacks. Different upper letters mean differences between collection periods, for each of the different snacks. Significant for *p*-value < 0.05.

**Figure 4 molecules-26-02403-f004:**
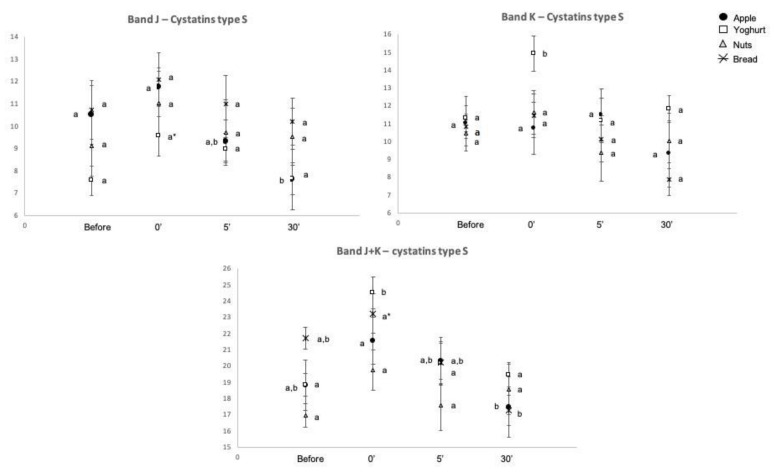
Variations in salivary cystatins induced by the intake of different snacks. Different upper letters mean differences between collection period, for each of the different snacks for *p*-value < 0.05; Upper letters with * mean differences between collection periods for 0.05 < *p*-value < 0.1.

**Figure 5 molecules-26-02403-f005:**
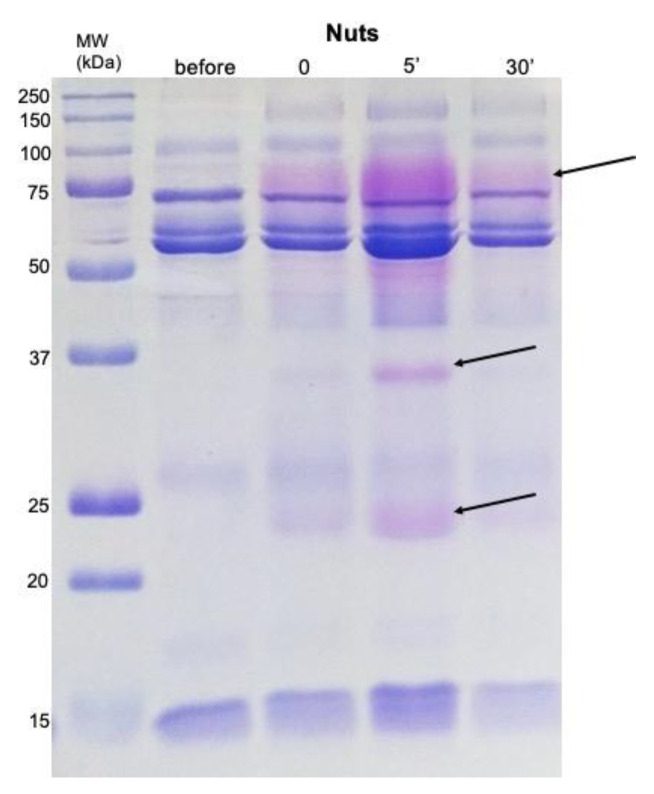
SDS-PAGE gel stained flowing Beeley’s protocol [25] for differential coloration of PRPs (arrows). Samples obtained from walnuts snack are presented as example.

**Table 1 molecules-26-02403-t001:** Variations in saliva composition through time, after intake of the 4 different snacks (means and SE): a two-way general linear model.

Time	Band A	Band B	Band C	Band D	Band E	Band F	Band G	Band H	Band I	Band J	Band K	Band J + K	Flow Rate (mL/min)	Total Protein (µg/mL)	Amy_ Enzymatic Activity ^1^ (U/mL)
**Before**	7.3(0.2) ^A^	7.9(0.4) ^A^	9.5(0.5)	14.0(0.9)	10.2(0.5)	2.5(0.3) ^A^	9.5(0.6) ^A^	4.9(0.4)	6.5(0.4)	9.3(0.6) ^A^	10.6(0.5) ^A^	18.9(0.8) ^A^	0.54(0.03) ^A^	670.7(38.9) ^A,B^	148.2(13.7) ^A^
**0′**	5.3(0.2) ^B^	8.1(0.4) ^A^	8.7(0.4)	14.8(0.6)	10.2(0.5)	4.0(0.4) ^B^	7.4(0.3) ^B^	5.6(0.7)	7.5(0.3)	11.1(0.6) ^B^	12.1(0.6) ^B^	22.1(0.8) ^B^	0.67(0.04) ^B^	715.5(38.7) ^A^	202.2(14.3) ^B^
**5′ after**	6.2(0.2) ^C^	9.2(0.5) ^B^	9.5(0.4)	14.8(0.8)	10.4(0.5)	2.8(0.3) ^A^	9.0(0.4) ^A^	4.8(0.4)	6.9(0.3)	9.8(0.6) ^A,B^	10.6(0.7) ^A,B^	19.5(1.0) ^A^	0.64(0.03) ^A^	576.5(35.2) ^A,B^	155.8(16.3) ^A^
**30′ after**	7.1(0.2) ^A^	8.2(0.4) ^A^	9.7(0.5)	15.0(0.9)	9.2(0.5)	2.6(0.3) ^A^	10.1(0.5) ^A^	5.5(0.5)	6.9(0.2)	8.7(0.6) ^A^	9.8(0.6) ^A^	18.2(0.9) ^A^	0.62(0.04) ^A^	566.9(46.1) ^B^	160.3(15.6) ^A,B^
*p*-value (effects)															
**Time**	0.0005 *	0.012 *	0.151	0.636	0.918	0.005 *	0.0005 *	0.437	0.176	0.025 *	0.014 *	0.0005 *	0.007 *	0.006 *	0.007 *
**Time * Snack**	0.273	0.0005 *	0.909	0.243	0.747	0.298	0.684	0.498	0.833	0.539	0.114	0.224	0.031 *	0.130	0.006 *

SE, standard error; Different upper letters represent differences among the different periods (* *p* < 0.05; Two-way ANOVA; Bonferrroni’s post-test); ^1^ Salivary amylase enzymatic activity.

**Table 2 molecules-26-02403-t002:** Mass spectrometry identification of salivary proteins present in the SDS-PAGE bands.

Protein Band	Protein ID	Accession Number (Uniprot)	MW (kDa) (Est/Theor.) #	MASCOT ID Score	N° Peptides Matched
A	Mixture (Polymeric Immunoglobulin receptor + Lactotransferrin)	P01833 + P02788	102.0/84.4 and 80.0	164	24
B	Albumin	P02768	71.0/71.3	150	15
C	Alpha-amylase 1	P04745	62.0/58.4	154	16
D	Alpha-amylase 1	P04745	58.0/58.4	135	15
E	Mixture (Carbonic Anhydrase VI + Zinc-alpha-2-glycoprotein)	P25311 + P23280	42.0/34.5 and 35.4	196	20
F	n.i.	___	___	___	___
G	Ig Kappa chain C region	P01834		71	4
H	n.i	___	___	___	___
I	Prolactin-inducible protein	P12273	17.0/16.8	92	7
J + K *	Cystatin SN	P01037	14.0/16.6	99	7

* Although image analysis allowed the analysis of bands J and K separately, for protein identification, given the close relationship between the bands, it was not possible to isolate them, so the identification was made for all this gel region. # Estimated and theoretical molecular masses. n.i., not identified.

## Data Availability

The data presented in this study are available on request from the corresponding author. All data relevant to the study are included in the article and access to raw data would be provided upon request.

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
