# Peer review of "How Different Snacks Produce a Distinct Effect in Salivary Protein Composition"

_molecules, 2021, doi:10.3390/molecules26092403_

Round 1

Reviewer 1 Report

The manuscript is well structured with a purpose, hypothesis, and results that are carefully discussed with other reports. The conclusions summarize the work. The Authors are aware of the research carried out, its advantages and weaknesses, and outline the direction of future analyzes.

However,  the reviewer expresses doubts as to the small number of candidates examined. And, what is the impact of research on the field of science? What do the obtained results contribute to the science development? What exactly do they bring?

Manuscript needs some editing corrections. Table 1 is doubled. Please improve the quality of figures and adapt tables, text layout and references to MDPI requirements (font, size, style, etc.).

Author Response

The manuscript is well structured with a purpose, hypothesis, and results that are carefully discussed with other reports. The conclusions summarize the work. The Authors are aware of the research carried out, its advantages and weaknesses, and outline the direction of future analyzes.

Authors - The authors would like to thank reviewer kind comments.

However, the reviewer expresses doubts as to the small number of candidates examined. And, what is the impact of research on the field of science? What do the obtained results contribute to the science development? What exactly do they bring?

Authors - We understand the reviewer concern about the number of individuals examined, but since 4 collections were required per participant and since each sample was run in duplicate, to minimize errors due to gel-to-gel variations, to work with a high number of individuals is not viable. Because of this, we opted by selecting a homogeneous sample, only constituted by women, of limited age range and having exactly the same food consumption anticipating saliva collection. This is now highlighted in the conclusions section of the revised version.

Manuscript needs some editing corrections. Table 1 is doubled.

Authors  - Corrected

Please improve the quality of figures and adapt tables, text layout and references to MDPI requirements (font, size, style, etc.).

Authors - Concerning tables, we need support from MDPI office to format Table 1.

Reviewer 2 Report

Simões et al present a study of changes in human salivary composition in response to eating several snacks. Overall it is well-written, and likely to be of interest to the readers of Molecules. In my opinion there are some edits and clarifications required to make this manuscript suitable for publication.

  1. Could the authors say when the data was collected? If after 2019, were participants tested for covid-19, and were changes made to the standard data collection setup to minimise transmission risk? It is of course fine if this is the case, but will aid reproducibility if details are given.

  1. Can the authors explain why they chose to give their participants coffee prior to the experiment, given potential effects of caffeine on salivation (e.g. DOI:10.4172/OHCR.1000107)?

  1. The exclusion criteria seem quite broad. Only female participants were used, but the authors do not include any references demonstrating sex-differences in salivation. These references should be included. Furthermore, excluding smokers would remove about 30% of the population in Portugal, and excluding users of any medication whatsoever would likely exclude a similar proportion. Finally, the age restriction (18-30) excludes the majority of the population. Could the authors add a few sentences to explain these choices?

  1. In figure 1, it is not entirely clear what “period 1” to “period 4” refers to. Are these the same as the “before, 0, 5, 30” labels on subsequent figures? Can this be made consistent between figure to aid the reader?

  1. In figure 1B, there is a huge difference in the protein concentration even in “period 1”, which I assume is before eating the snack (~500 ug/ml for bread compared to ~800 ug/ml for apple). This does not make sense, and to me, brings the much smaller snack-induced changes into question. However, the authors do not discuss this at all.

  1. Based on figure 1B, the authors say that the changes for yoghurt are significantly different, whereas the changes for walnuts are not. This is despite the fact that the yoghurt and walnut timeseries are almost identical. Again, this does not make sense, but is not discussed by the authors.

  1. Figures 3 and 4 contain numerous markings of “a”, “b” and “a,b”, but the authors do not explain what these mean.

  1. In my view, the use of “tendency” to claim a difference when the statistical analysis does not give the desired result is inappropriate, and should be removed throughout.

  1. Line 223: “With the exception of walnuts intake, the levels of immunoglobulin chains decreased… suggesting that this salivary response to mastication is not food-specific.“ This is confusing. Surely the authors’ result from walnuts suggest that it IS food-specific? This needs to be clarified.

Author Response

Simões et al present a study of changes in human salivary composition in response to eating several snacks. Overall it is well-written, and likely to be of interest to the readers of Molecules. In my opinion there are some edits and clarifications required to make this manuscript suitable for publication.

Authors - We would like the reviewer for the comments, and we corrected the manuscript to address all the concerns. We hope we could improve it, with the changes made.

  1. Could the authors say when the data was collected? If after 2019, were participants tested for covid-19, and were changes made to the standard data collection setup to minimise transmission risk? It is of course fine if this is the case, but will aid reproducibility if details are given.

Authors - Data was collected in February-March 2019. This information was introduced in material and methods section.

  1. Can the authors explain why they chose to give their participants coffee prior to the experiment, given potential effects of caffeine on salivation (e.g. DOI:10.4172/OHCR.1000107)?

Authors - We thank the reviewer question, which is pertinent. Coffee was considered because we wanted to test individuals in a situation similar to the one they are when consuming snacks (in this case in the midle of the morning). As such, we opted by give them the type of breakfast they usually consume and coffee was reported to be consumed daily after breakfast by participants. Coffee can induce changes in saliva composition. However, coffee was consumed immediately after breakfast, 1h30m before experiment, so we do not expect a major effect. But even being possible to have it at that time, all participants consumed the same amount of coffee, at the same time, in the different days of the experiment. As such, it is not expected that coffee would influence the comparisons between snacks, for their effect in saliva.

The reason why coffee was considered, after breakfast, was added to material and methods.

  1. The exclusion criteria seem quite broad. Only female participants were used, but the authors do not include any references demonstrating sex-differences in salivation. These references should be included. Furthermore, excluding smokers would remove about 30% of the population in Portugal, and excluding users of any medication whatsoever would likely exclude a similar proportion. Finally, the age restriction (18-30) excludes the majority of the population. Could the authors add a few sentences to explain these choices?

Authors - The exclusion criteria aimed to avoid having factors that may affect salivatory function (and consequently the capacity of salivatory response to food stimuli) and that, at the same time, would differ among individuals (e.g. the existence or the type of medication, to be, or not smoker and the level of tobacco use in the ones smoking, etc.). Moreover, only normal-weight individuals were studied, since BMI also affects salivation (that was not referred previously but was now added to this revised version). Concerning the age, young adults were chosen to work with adults, but at the same time avoid to extend to a high range of ages, that could increase variability. All this was particularly important, since we worked with a limited number of participants, due to the limitation of the approach used: gel-based proteomics approaches.

Authors - Sentences explaining the reason for the exclusion criteria was added in discussion section, as suggested.

  1. In figure 1, it is not entirely clear what “period 1” to “period 4” refers to. Are these the same as the “before, 0, 5, 30” labels on subsequent figures? Can this be made consistent between figure to aid the reader?

Authors - Figure 1 was corrected accordingly.

  1. In figure 1B, there is a huge difference in the protein concentration even in “period 1”, which I assume is before eating the snack (~500 ug/ml for bread compared to ~800 ug/ml for apple). This does not make sense, and to me, brings the much smaller snack-induced changes into question. However, the authors do not discuss this at all.

Authors - This is because salivary parameters are variable among different individuals, being less variable for the same individual. That is why the design used, where each individual was control of itself, is adequate for this type of sample. This is now discussed in the revised version.

  1. Based on figure 1B, the authors say that the changes for yoghurt are significantly different, whereas the changes for walnuts are not. This is despite the fact that the yoghurt and walnut timeseries are almost identical. Again, this does not make sense, but is not discussed by the authors.

Authors - This lack of statistically significant difference is due to the high variability in total concentration values. This is now discussed, in this revised version.

  1. Figures 3 and 4 contain numerous markings of “a”, “b” and “a,b”, but the authors do not explain what these mean.

Authors - Information that different upper letters mean differences between periods, for each snack, is present in Fig 3 and Fig 4 legends.

  1. In my view, the use of “tendency” to claim a difference when the statistical analysis does not give the desired result is inappropriate, and should be removed throughout.

Authors - We agree with the reviewer and corrected the manuscript accordingly.

  1. Line 223: “With the exception of walnuts intake, the levels of immunoglobulin chains decreased… suggesting that this salivary response to mastication is not food-specific.“ This is confusing. Surely the authors’ result from walnuts suggest that it IS food-specific? This needs to be clarified.

Authors - We thank the reviewer for the repair. What we want to pass is that the decrease in the levels of salivary Ig can be more related with chewing than with the food chemical characteristics. In the case of walnuts, lower mastication was required, since to reach the same amount of energy, the total amount of nuts was lower than the amount of other snacks. The entire paragraph was re-written to clarify that chewing may have a role in this change observed for salivary Ig.